# Group Fairness in Predict-Then-Optimize Settings for Restless Bandits

**Shresth Verma**[*1]    **Yunfan Zhao**[*1]    **Sanket Shah**[1]    **Niclas Boehmer**[1]    **Aparna Taneja**[2]    **Milind Tambe**[1,2]

[1]Harvard University
[2]Google Research India

## Abstract

Restless multi-arm bandits (RMABs) are a model for sequentially allocating a limited number of resources to agents modeled as Markov Decision Processes. RMABs have applications in cellular networks, anti-poaching, and in particular, healthcare. For such high-stakes use cases, allocations are often required to treat different groups of agents (e.g., defined by sensitive attributes) fairly. In addition to the fairness challenge, agents' transition probabilities are often unknown and need to be learned in real-world problems. Thus, group fairness in RMABs requires us to simultaneously learn transition probabilities and how much budget we allocate to each group. Overcoming this key challenge ignored by previous work, we develop a decision-focused-learning pipeline to solve *equitable RMABs*, using a novel budget allocation algorithm to prevent disparity between groups. Our results on both synthetic and real-world large-scale datasets demonstrate that incorporating fair planning into the learning step greatly improves equity with little sacrifice in utility.

## 1 INTRODUCTION

Restless multi-arm bandits (RMABs) are a model for sequentially distributing scarce resources to a set of agents. Concretely, we have a set of arms and a limited budget and face the question of deciding which arms to pull in each round. The state of arms evolves according to a Markov Decision Process where transition probabilities depend on whether the arm is pulled in this step. RMABs have a broad range of applications, including resource allocation in anti-poaching, machine maintenance, cellular networks [Modi et al., 2019, Zhao et al., 2008, Bagheri and Scaglione, 2015,

Glazebrook et al., 2006, Qian et al., 2016, Yu et al., 2018, Ruiz-Hernández et al., 2020]. RMABs have especially be used in healthcare settings such as call scheduling in a maternal and child care program [Mate et al., 2022, Killian et al., 2023], screening patients at risk of cancer [Lee et al., 2019], and allocating hepatitis C treatment [Ayer et al., 2019].

In high-stakes resource allocation settings such as healthcare, authorities often want to give priority to agent groups, e.g., as defined by sensitive attributes, that are most in need. For instance, some governments require non-discrimination based on sensitive attributes [Amon, 2020] and non-profits commonly aim to prioritize low income groups [Verma et al., 2023b]. The standard RMAB objective of maximizing arms' summed reward falls short in these cases. For example, if one unit of budget could increase the utility of a marginalized group from 0.2 to 0.35 or increase that of a non-marginalized group from 0.5 to 0.7, a strictly utilitarian policy would favor the non-marginalized group, further widening the socio-economic gap. Importantly, equal allocation is often not sufficient, and equity or balanced outcomes is preferred [Marsh and Schilling, 1994, Luss, 2012]

While the literature [Herlihy et al., 2023, Li and Varakantham, 2022, Killian et al., 2023] has mainly focused on how to plan fairly, a main challenge in using RMABs in the real world is how to predict unknown transition probabilities of arms. The task of predicting transition probabilities that are later used in a planning problem falls within a well-studied predict-then-optimize framework [Elmachtoub and Grigas, 2022]. A naive "two-stage" approach is to first learn transition probabilities minimizing prediction error and subsequently use them to decide on an allocation maximizing the fairness objective. However, it has been shown that such approaches suffer from objective mismatch [Wilder et al., 2019] and inaccurate predictions in our setting may even increase disparity among groups.

Decision-focused-learning (DFL) approaches that incorporate the downstream optimization problem have proven to be both scalable and effective [Mandi et al., 2023, Agrawal

---
[*]Equation Contribution

et al., 2019], and have been studied for RMABs when maximizing the summed reward [Wang et al., 2023]. However, applying the DFL paradigm to optimize a group fairness objective presents us with novel challenges: The optimization objective becomes more complex and makes it necessary to simultaneously learn agents' transition probabilities and the allocation of budget to groups. To incorporate how both transition probabilities and budget allocations affect the fairness objective, one needs to differentiate through the process of allocating budgets to different groups.

Tackling this challenge, we provide a DFL pipeline for *equitable* RMABs that simultaneously learns budget allocation and transition probabilities, in an offline learning setting. Our main contributions are:

- To the best of our knowledge, we are the first to develop a decision-focused-learning (DFL) method to solve *equitable* RMABs, outperforming various baselines by 20-40% in fairness objectives, on both synthetic and real-world large-scale datasets.

- We propose a novel differentiable budget allocation method such that the effect of changes in budget on the fairness objective is incorporated during training. Our theoretical results shed light on the feasibility and effectiveness of this method.

- Our method optimizes fairness objectives while achieving utility $>90\%$ of a state-of-the-art algorithm [Wang et al., 2023] that purely optimizes utility, on a real-world health information dataset.

- Our pipeline is compatible with a broad range of fairness objectives, including Maximin Reward, Gini Index, and Max Nash Welfare, and generally to a range of functions defined over utilities of groups.

## 2 LITERATURE REVIEW

**RMAB** Restless multi-arm bandits are shown to be PSPACE hard [Papadimitriou and Tsitsiklis, 1994] and approximated algorithms have been proposed [Hawkins, 2003, Whittle, 1988]. In particular, Whittle [1988] uses a Lagrangian relaxation to decouple computations for arms and select actions by computing Whittle indices of each arm. The resulting Whittle index policy is asymptotically optimal under an indexability assumption [Akbarzadeh and Mahajan, 2019, Weber and Weiss, 1990].

**Fairness in RMABs** Li and Varakantham [2022], Herlihy et al. [2023] consider individual fairness constraints that ensure sufficient resources are given to each arm. Li and Varakantham [2023] studies fairness by always probabilistically favoring an arm that yields higher long-term cumulative reward. Biswas et al. [2023] studies fairness for workers who pull the arms, ensuring that we never overburden the workers. However, above works do not consider group-level

fairness, where balancing group outcomes is a key challenge. Killian et al. [2023] designs RMAB algorithms to achieve equal outcomes but fails to address practical settings with unknown transition dynamics. Notice none of the above works have dealt with unknown transition probabilities.

**Decision focused learning** The predict-then-optimize framework contains a prediction problem where predicted parameters are used in a downstream optimization problem that solves for a solution, and the overall goal is to obtain high-quality solutions [Elmachtoub and Grigas, 2022]. Two-stage approaches that ignore the downstream optimization problem when solving the prediction problem result in a mismatch of the prediction loss and the solution quality [Lambert et al., 2020, Wilder et al., 2019, Mandi et al., 2020]. Incorporating the downstream optimization task in the prediction problem is shown to improve model performance both theoretically [Grigas et al., 2021, Mandi et al., 2023, Shah et al., 2022] and empirically [Amos and Kolter, 2017, Huang et al., 2019, Agrawal et al., 2019, Wang et al., 2021]. Verma et al. [2023a] and Wang et al. [2023] provide DFL methods tailored to RMABs but ignore fairness.

**Fair top-k ranking** Choosing which arms to pull is conceptually similar to top-k ranking. Singh and Joachims [2019], Yadav et al. [2021] provide policy gradient methods for learning fair ranking policies. Celis et al. [2017] constructs a mixed integer programming problem to solve for rankings and explicitly enforces fair representation between groups. Zehlike et al. [2017] studies fairness up to a threshold and provides a greedy algorithm to achieve fair ranking. Singh and Joachims [2018], Biega et al. [2018] use fairness constraints that link relevance to allocation to exposure items. Interestingly, top-k selection can be formulated as an LP and solved using DFL [Kotary et al., 2022, Wilder et al., 2019]. However, existing works in fair top-k ranking do not address MDPs, where actions affect future states.

**Equity and Group Fairness** Since groups have different needs, allocating the same resources to each group is not sufficient, and balanced outcomes are preferred [Luss, 2012, McGlaughlin and Garg, 2020]. Although much theory in social welfare studies individual level equity, equity is often measured on a group level in applications such as facility location [Marsh and Schilling, 1994], healthcare [Braveman and Gruskin, 2003], and humanitarian logistics [Gutjahr and Fischer, 2018]. Existing works on fair allocation extend equity to a group level [Barman et al., 2018, Suksompong, 2018] but fail to consider settings where resources are scarce and only a small subpopulation receives resources.

# 3 PRELIMINARIES

## 3.1 RMABS AND THE WHITTLE INDEX POLICY

We consider a restless multi-arm bandit problem with $N$ arms and a budget of $B$. We focus on an offline learning setting where a historical dataset is available. Each arm has a discrete state space $\mathcal{S}_i$ and actions are binary $\{0,1\}$. We denote the transition probability of arm $i$ from state $s$ to state $s'$ under action $a$ as $P_i(s,a,s')$. The transition probabilities are unknown and arm features $x_i$ are available. We let $R_i(s)$ denote the per-arm reward for arm $i$ at state $s$. Since in practice, it is common that $S_i, R_i$ are the same for all arms [Herlihy et al., 2023, Mate et al., 2022], we drop the subscript $i$. Notice our methods apply to the general setting. We let $M = |\mathcal{S}|$ denote the number of possible states. The vector of arm states is $s \in \mathbb{R}^N$, and the one-hot encoding of actions on arms is $a \in \{0,1\}^N$. A policy $\pi$ maps arm states $s$ to actions $a$, while satisfying budget constraints $\sum_{i=1}^N a_i \leq B, \forall t$ at each timestep $t \in [H]$. For a reward-maximizing RMAB without fairness considerations, the optimal policy maximizes the following Bellman equation:

$$V(s,B) = \max_{a} \left\{ \sum_{i=1}^N R(s_i) + \beta \mathbb{E}\left[V(s',B) \mid s,a\right] \right\},$$

$$\text{s.t. } \sum_{i=1}^N a_i \leq B, \tag{1}$$

where $\beta \in (0,1]$ is a discount factor. Note the state space and the action space grow exponentially in $N$. To learn a policy, a scalable approach is to use the Whittle Index derived from the Lagrangian relaxation [Whittle, 1988].

$$J(s,B) = \min_{\lambda \geq 0} \left( \frac{\lambda B}{1-\beta} + \sum_{i=1}^N \max_{a_i \in \{0,1\}} \left\{ Q_i(s_i, a_i, \lambda) \right\} \right),$$
$$\tag{2}$$

$$\text{s.t. } Q_i(s_i, a_i, \lambda) = R(s_i) - \lambda a_i + \beta \mathbb{E}\left[Q_i(s_i', a_i, \lambda) \mid \pi(\lambda)\right].$$

The Whittle Index $W_i(s_i)$ is equal to the smallest action charge $m$ that makes pulling as rewarding as not pulling

**Definition 1** (Whittle index). *The Whittle index associated to state $s_i$ is:*

$$W_i(s_i) := \inf_m \left\{ Q_i(s_i, a_i = 0, m) = Q_i(s_i, a_i = 1, m) \right\}.$$

Intuitively, given a unit budget, an arm with a higher Whittle index would benefit from a larger increase in discounted cumulative reward. The Whittle Index policy $\pi^{\text{whittle}}$ is to pull $B$ arms with the highest Whittle indices.

## 3.2 FAIRNESS OBJECTIVES AND GROUP BUDGET ALLOCATIONS

The set of $N$ arms can be partitioned into groups $\mathcal{G}$ according to known arm features (such as age, geographical location, and education [Mate et al., 2022]). Let $N_g$ denote the size of group $g$. We define the value function for a group:

$$V_g(s_g, b_g) = \frac{1}{N_g} V(s_g, b_g), \tag{3}$$

where $s_g \in \{0,1\}^{N_g}$ is a vector representing states of all arms in group $g$. We develop a differentiable pipeline that accommodates any objective that satisfies a key assumption:

**Assumption 1.** *The fairness objective or a proxy of it is differentiable in group values (3).*

A fairness objective that fails to satisfy Assumption 1 has little hope of being compatible with a differentiable pipeline, where the gradient of the objective with respect to group budget allocations needs to be evaluated. Many popular fairness objectives, including Max Nash Welfare (MNW), Maximin Reward (MMR), and Gini Index, satisfy Assumption 1. Given fixed groups $g \in \mathcal{G}$, MNW optimizes the product of group values:

$$\text{MNW}(s) := \max_{b_g, g \in \mathcal{G}} \prod_{g \in \mathcal{G}} V_g(s_g, b_g) \text{ s.t. } \sum_{g \in \mathcal{G}} b_g = B. \tag{4}$$

By giving diminishing returns for any group's marginal increase in utility, MNW naturally trades off efficiency and equity [Caragiannis et al., 2019, Ramezani and Endriss, 2009]. MMR optimizes the so-called egalitarian social welfare [Asadpour and Saberi, 2007, Caragiannis et al., 2012]:

$$\text{MMR}(s) := \max_{b_g, g \in \mathcal{G}} \min_{g \in \mathcal{G}} V_g(s_g, b_g) \text{ s.t. } \sum_{g \in \mathcal{G}} b_g = B. \tag{5}$$

By maximizing the utility of the worst-off group, MMR is shown to be favorable both theoretically and empirically [Brandt et al., 2016, Bonald et al., 2006]. We will provide a differentiable proxy of MMR.

# 4 EQUITABLE RMABS WITH DECISION-FOCUSED-LEARNING

We begin by illustrating that naive approaches to address fairness fail. Next, we propose a differentiable pipeline compatible with a broad range of fairness objectives and provide theoretical results that shed light on the feasibility and effectiveness of our pipeline. After that, we discuss details on how to use the pipeline with various fairness objectives.

## 4.1 NEED FOR A DIFFERENTIABLE PIPELINE

One naive approach is to prioritize learning optimal transition probabilities to maximize utility, and then allocate budgets proportional to group size. This approach does not simultaneously learn budget allocations and transition probabilities, and thus it does not take into account groups'

**Algorithm 1** Greedy Budget Allocation

**Input**: Sorted Whittle indices $W$, groups $g \in \mathcal{G}$, budget $B$, states $\boldsymbol{s}$, initial budgets $b_g = 0, \forall g$

1: Compute for each group $g \in \mathcal{G}$,
   $J_\Delta(\boldsymbol{s}_g, b_g) = \log(J(\boldsymbol{s}_g, b_g + 1)) - \log(J(\boldsymbol{s}_g, b_g))$
2: **while** there is budget remaining **do**
3:     Give one more budget to the group with the highest $J_\Delta(s_g, b_g)$ and recompute $J_\Delta(s_g, b_g)$ for this group.
4: **return** budget allocation $b_g$

---

needs. We demonstrate it fails to provide sufficient budgets to marginalized groups (see Figure 3).

A more principled approach is to first predict the transition probabilities by minimizing a prediction loss, and then plugin the predictions into a budget allocation algorithm in Killian et al. [2023] that maximizes the fairness objective. However, predictions from such a two-stage procedure can be of poor quality [Wang et al., 2023], which leads to poor budget allocations (see Figure 1).

### 4.1.1 A Non-Differentiable Approach

Having discussed why alternatives fail, we introduce our first DFL approach that simultaneously allocates budgets and predicts transition probabilities. We will first describe a budget allocation subroutine, and then discuss a main algorithm that uses the subroutine. We explain this approach using MNW, and later discuss other fairness objectives.

**Budget Allocation** Problem (4) can be rewritten as:

$$\max_{b_g} \sum_{g \in \mathcal{G}} \log(V_g(s_g, b_g)) \quad \text{s.t.} \sum_{g \in \mathcal{G}} b_g = B. \quad (6)$$

Since computing $V_g(s_g, b_g)$ is PSPACE hard [Papadimitriou and Tsitsiklis, 1994], following Killian et al. [2023], we replace $V_g(s_g, b_g)$ by the Lagrangian relaxation $J(s_g, b_g)$ that upper bounds it. Taking $\frac{dJ(s_g, b_g)}{db_g} = \frac{\lambda}{1-\beta}$, we have that $J(s_g, b_g)$ is increasing in $b_g$. In addition, when $b_g$ increases, an optimal policy would take more actions, which implies a lower action charge $\lambda_k$. Thus, $\frac{dJ(s_g, b_g)}{db_g} = \frac{\lambda}{1-\beta}$ is decreasing in $b_g$, and $J(s_g, b_g)$ is concave. Consequently, to maximize MNW, it suffices to greedily assign one additional budget to the group that achieves the maximum increase in $\log J(s_g, b_g)$. Based on above, Algorithm 1 gives budget to the group that provides the highest increase in log MNW.

**Main Algorithm** Algorithm 2 simultaneously predicts transition probabilities and allocates group budgets. At each epoch, we predict transition probabilities and compute whittle indices (lines 3-4). After that, we allocate budgets (lines 5-6) and collect trajectories using the Whittle index policy $\pi^{\text{whittle}}$ (lines 7-9). For each group, having the budget allocation, we choose arms to pull according to a soft version of

---

**Algorithm 2** Equitable DFL Algorithm for RMABs

**Input**: offline dataset, groups $g \in \mathcal{G}$, learning rate $\alpha_w$, frequency $n_f$, warm-up epochs $n_{ini}$

1: Initialize a neural network with weights $w$ to predict transition probabilities
2: **for** epoch = 1,2,... **do**
3:     Predict transition probabilities $P$
4:     Compute Whittle indices $W$.
5:     **if** $epoch > n_{ini}$ and $(epoch - n_{ini})\% n_f == 0$ **then**
6:        Allocate budget $b_g$ using Algorithm 1 or 3
7:     **for** timestep t =1,2,...,H **do**
8:        **for** $g \in \mathcal{G}$ **do**
9:           Given $b_g$, Take actions according to the Whittle index policy $\pi^{\text{Whittle}}$ with soft-top-$B$ selection.
10:     For each group $g$, use importance sampling to compute group value $V_g(s_g, b_g)$.
11:     Compute Max Nash Welfare objective
    $\text{MNW} = \prod_g V_g(s_g, b_g)$
12:     Update $w \leftarrow w + \alpha_w \frac{d\,\text{MNW}}{d\pi^{\text{Whittle}}} \frac{d\pi^{\text{Whittle}}}{dW} \frac{dW}{dP} \frac{dP}{dw}$
13: **return** trained neural network with weights $w$

---

Whittle index policy, where instead of pulling arms with top Whittle indices, we use a soft-top-$K$ selection [Xie et al., 2020]. Using the collected trajectories and importance sampling, we compute the Max Nash Welfare objective (lines 11). Finally, we update the weights using the gradient. See Appendix A for additional details, including importance sampling (line 10) and gradient computations (line 12).

**Limitations.** Our first approach already addresses limitations of simple alternatives and provides substantial improvement in the MNW objective in several settings (see Figure 1). However, this approach does not consistently outperform, and it fails to allocate sufficient budgets to a marginalized group (see Figure 3). A key limitation of Algorithm 1 is that the budget allocation is non-differentiable and relies heavily on estimates of $J_\Delta(\boldsymbol{s}_g, b_g)$. When transition probability predictions are poor, $J_\Delta(\boldsymbol{s}_g, b_g)$ estimates are inaccurate, and the resulting poor budget allocations in turn harm transition probability learning. In addition, the first approach requires different procedures tailored to distinct fairness objectives (e.g. for a fairness objective other than MNW, we need to provide another greedy procedure based on its properties).

### 4.2 OUR DIFFERENTIABLE PIPELINE

To address limitations in the first approach, we propose to use a differentiable budget allocation procedure (Algorithm 3), where we update budget allocations by taking gradient steps. To use this procedure, in the main algorithm (Algorithm 2), when allocating budgets (line 6), we call the differentiable procedure (Algorithm 3).

**Algorithm 3** Differentiable Budget Allocation

---

**Input**: groups $g \in \mathcal{G}$, budget $B$, learning rate $\alpha_b$, (if not the first time this algorithm is called: previous $b_g$, $\pi^{\text{Whittle}}$, and $MNW$)

1: **if** $1^{st}$ time this algorithm is called **then**
2:   Initialize budgets $b_g$ proportional to group size.
3: **else**
4:   Update $b_g \leftarrow b_g + \alpha_b \frac{d\,\text{MNW}}{d\pi^{\text{Whittle}}} \frac{d\pi^{\text{Whittle}}}{db_g}$
5: **return** budget allocation $b_g$

---

We explain the pipeline using MNW and later discuss other fairness objectives. The gradient of the MNW objective with respect to $b_g$ can be calculated as $\frac{d\,\text{MNW}}{d\pi^{\text{Whittle}}} \frac{d\pi^{\text{Whittle}}}{db_g}$. To evaluate the first term, the gradient of the MNW objective with respect to the Whittle indices, we use the policy gradient Theorem [Sutton et al., 1998]. To calculate the second term, we establish the following proposition:

**Proposition 1.** *For each group $g \in \mathcal{G}$, to compute an approximate of the gradient $\frac{d\pi^{\text{Whittle}}}{db_g}$, it is sufficient to know Whittle indices of arms in $g$, budget $b_g$, the number of arms $N_g$ in group $g$.*

*Proof of Proposition 1.* Pulling top-$b_g$ arms from each group $g \in \mathcal{G}$ can be formulated as an optimal transport problem. Specifically, we let $\boldsymbol{\mu} := \frac{\mathbf{1}_{N_g}}{N_g} \in \mathbb{R}^{N_g}$ and $\boldsymbol{v} := [\frac{b_g}{N_g}, \frac{N_g - b_g}{N_g}]$. We let $y := [0,1]^\top$ and a cost matrix $M_{ij} := |\bar{W}_i - y_j|^2$, where $\bar{W}_i$ is normalized Whittle index of arm $i \in g$. The top-$b_g$ operator output can be obtained from a linear mapping of the optimal transport plan $T^*$ [Xie et al., 2020]:

$$S(\boldsymbol{\mu}, \boldsymbol{v}) := \min_{T \in \Pi(\boldsymbol{\mu}, \boldsymbol{v})} \langle T, M \rangle, \qquad (7)$$

where $\Pi(\boldsymbol{\mu}, \boldsymbol{v}) := \{T \in \mathbb{R}_+^{N_g, 2} | T\mathbf{1}_2 = \boldsymbol{\mu}, T\mathbf{1}_{N_g} = \boldsymbol{v}\}$. Notice solving the optimization problem (7) only requires that we know $\bar{W}_i$, $b_g$, and $N_g$.

Solving the optimization problem (7) is expensive and a regularized version is commonly used [Cuturi, 2013]:

$$\tilde{S}_\epsilon(\boldsymbol{\mu}, \boldsymbol{v}) := \min_{T \in \Pi(\boldsymbol{\mu}, \boldsymbol{v})} \langle T, M \rangle + \epsilon \sum_{i,j} T_{ij}(\log T_{ij} - 1).$$

Using the regularized version, an approximate gradient of the objective in (7) with respect to input $\boldsymbol{v}$ can be computed (Algorithm 1 in Luise et al. [2018]). $\square$

Note the proof of Proposition 1 already provides detailed procedures on gradient computations. When performing gradient updates, we project the learned budget onto the feasible region (we apply softmax normalization in the last layer of the neural network learning per group budget allocations). Thus, the budget constraint is satisfied throughout.

Since our pipeline relies on gradient updates, it would be desirable if the optimization landscape is "nice". Specifically, we will argue that a proxy of MNW that extends to non-integer values is concave. The analysis, tailored to RMAB problems, requires arguments to address that budgets are usually integer-valued. Observe that solving MNW (Problem 4) is equivalent to solving the log of MNW (Problem 6). For ease of exposition, we define under a fixed budget $b_g$ of a given group $g$:

$$h_g^{MNW}(b_g) := \max_{\{b_{g'}\}_{g' \in \mathcal{G} \setminus g}} \sum_{g' \in \mathcal{G}} \log V_g(s_g, b_g) \text{ s.t.} \sum_{g' \in \mathcal{G}} = B.$$

Since we cannot pull 0.5 arm and $h_g^{MNW}(b_g)$ is only defined on integer valued points $b_g \in \mathbb{Z}$, we construct a proxy of $h_g^{MNW}(\cdot)$ that is defined on continuous values $b_g \in \mathbb{R}$:

$$\hat{h}_g^{MNW}(b_g) := h_g^{MNW}(\lfloor b_g \rfloor)$$
$$+ (b_g - \lfloor b_g \rfloor) \cdot (h_g^{MNW}(\lceil b_g \rceil) - h_g^{MNW}(\lfloor b_g \rfloor))$$

Observe that $\hat{h}_g^{MNW}(b_g) = h_g^{MNW}(b_g)$ for $b_g \in \mathbb{Z}$, and on non-integer valued $b_g$ the function $\hat{h}_g^{MNW}(\cdot)$ is a linear extrapolation based on nearest integer points. Additionally, $\hat{h}_g^{MNW}(\cdot)$ is piecewise linear. We now prove a result on the concavity of $\hat{h}_g^{MNW}(\cdot)$.

**Theorem 1.** *Assume $V_\Delta(b_g) := V_g(b_g + 1) - V_g(b_g)$ is a decreasing function of $b_g$ for each group $g$. For any group $g \in |\mathcal{G}|$, $\hat{h}_g^{MNW}(\cdot)$ is concave.*

To give an example, we consider a setting with two groups $\mathcal{G} = \{g_1, g_2\}$. Since the total budget is $B$, it suffices to choose $b_{g_1}$ and $b_{g_2} = B - b_{g_1}$ can be easily calculated. From Theorem 1, we have that $\hat{h}_{g_1}^{MNW}()$ is concave, implying that any local optima is a global optima. Thus, it suffices to start from any feasible integer values of $b_{g_1}$ and then iteratively move $b_{g_1}$ in steepest descent direction by one unit until convergence. Since $b_{g_1} \in [B]$ has at most $B + 1$ possible choices, following gradient updates (line 4, Algorithm 3), our procedure will terminate in finite amount of iterations.

**Complexity.** Algorithm 3 evaluates gradients separately for each group $g \in \mathcal{G}$ and each evaluation has complexity $O(N)$ [Luise et al., 2018], and thus the total cost is $O(N|\mathcal{G}|)$. In contrast, Algorithm 1 has complexity $O(N \log(N)|\mathcal{G}|BH)$, where $H$ is the length of the trajectory used to compute $J(s_g, b_g)$. Specifically, the factor $|\mathcal{G}|B$ is due to that we start with computing $J(s_g, b_g)$ for each group and then greedily allocate one budget at a time. The factor $N \log(N)H$ is the cost of computing $J(s_g, b_g)$ [Killian et al., 2023].

## 4.3 DIFFERENT FAIRNESS OBJECTIVES

While the first approach uses Algorithm 1 that heavily relies on the special properties of MNW and is not compatible with

other fairness objectives, our differentiable pipeline using Algorithm 3 accommodates various fairness objectives.

The MNW objective is a product (Equation 4), which is differentiable with respect to group value functions. The Maximin Reward (MMR) objective employs a minimum operator, which is not differentiable. To address that, we approximate the minimum operator with the Hölder mean:

$$f_p(x_1, ..., x_k) := \left( \frac{1}{k} \sum_i x_i^p \right)^{\frac{1}{p}}, \quad \text{for} \quad p \to -\infty$$

It is well-known that

$$\lim_{p \to -\infty} f_p(x_1, ..., x_k) = \min(x_1, ..., x_k).$$

Thus, replacing MNW in Algorithm 2 and 3 with $f_p(\cdot)$ applied on group values $V(s_g, b_g)$, we obtain a differentiable pipeline for MMR objective. Next, we analyze the optimization landscape for MMR, and argue a proxy of it that extends to non-integer values, is concave. Although the results are similar to that in MNW, the analysis tailored to MMR requires different arguments. We define under a fixed budget $b_g$ of a given group $g$:

$$h_g^{MMR}(b_g) := \max_{\{b_{g'}\}_{g' \in \mathcal{G} \setminus g}} \min_{g' \in \mathcal{G}} V_g(s_g, b_g) \text{ s.t.} \sum_{g' \in \mathcal{G}} = B,$$

and its proxy on continuous values $b_g \in \mathbb{R}$:

$$\hat{h}_g^{MMR}(b_g) := h_g^{MMR}(\lfloor b_g \rfloor)$$
$$+ (b_g - \lfloor b_g \rfloor) \cdot (h_g^{MMR}(\lceil b_g \rceil) - h_g^{MMR}(\lfloor b_g \rfloor))$$

We have the following result:

**Theorem 2.** *Assume $V_\Delta(b_g) := V_g(b_g + 1) - V_g(b_g)$ is a decreasing function of $b_g$ for each group $g$. For any group $g \in |\mathcal{G}|$, $\hat{h}_g^{MMR}(\cdot)$ is concave.*

We conduct experiments on MNW and MMR. Note our pipeline is compatible with any objective that satisfies Assumption 1, including Gini Index (see Appendix A).

# 5 EXPERIMENTS

We consider the following baselines. The parameters to learn are the transition probabilities of the RMAB problem. The Decision-focused-learning methods (DF) learn transition probabilities by maximizing the decision objective. The objective evaluated using importance sampling (see Appendix A), varies for distinct methods.

1. **Killian et al. [2023] (Two Stage Learning + Greedy Budget Allocation)**: Transition probabilities are learned by maximizing the predictive accuracy. Group budgets are computed greedily (Algorithm 1) where the Whittle index policy is applied for every group.

2. **DF-NoFair (DFL+No Fairness)**: The decision objective here is total utility. Group budgets are not computed and the Whittle Index policy is applied to the entire population. This method is equivalent to that described in Wang et al. [2023].

3. **DF-PropB (DFL+Proportional Budget)**: DFL decision objective here is total utility. We set group budgets proportional to group sizes and then apply the Whittle index policy for every group.

4. **DF-GreedyB (DFL+Greedy Budget Allocation)**: DFL decision objective here is a fairness metric. Group budgets are computed greedily (Algorithm 1) then apply Whittle index policy for every group.

5. **DF-LearnB (DFL+Learnable Budget Allocation)**: This is our differentiable pipeline. DFL decision objective here is a fairness metric. Group budgets are learned using Algorithm 3. The Whittle index policy is applied to every group.

We conduct experiments on both synthetic and real-world large-scale datasets, which we describe below.

**Synthetic** The synthetic dataset models an RMAB problem with 2 states $\{0, 1\}$ for $N$ arms, $B$ budget, time horizon $T = 10$, and a reward discount rate of $\gamma = 0.99$. For each arm and time step, we collect a unit reward at state 1 and 0 reward otherwise. We consider two settings with different group characteristics. **i) Two Groups**: one group is severely disadvantaged and there is a [50%, 50%] split between groups. **ii) Three groups** : The first group is severely disadvantaged and the second group is moderately disadvantaged. There is a [33%, 33%, 34%] split between groups. Compared to arms in the advantaged group, arms in disadvantaged groups obtain lower reward when not pulled but also obtain a lower increase in reward when pulled. Transition probabilities are randomly generated while enforcing that pulling is always strictly better than not pulling as well as the group characteristics. Having sampled the transition probabilities, we map them to features in $\mathbb{R}^{16}$ using a randomly initialized neural network. We study an offline problem with historical data collected by running a random behavioral policy.

**Real-world data** The dataset is collected by ARMMAN [ARMMAN, 2019], an NGO in India working on improving health awareness for expectant and new mothers. The program has enrolled over one million mothers, and health workers periodically make service calls to boost mothers' engagement in ARMMAN's health information program. Allocating limited service calls has been modeled as an RMAB problem with two actions (a health worker initiates a service call to the mother or not) and two states (engaging or not) [Mate et al., 2022, Verma et al., 2023b]. Each beneficiary is modeled as a Markov Decision Process, with unknown transition dynamics that can be inferred from known features. Each week, mothers are in the engaging

state if they listen to a health information voice message sent by ARMMAN for more than 30 seconds. Using a service quality improvement study of 44K mothers conducted by ARMMAN in January 2022 (see Appendix B.3.2 for data usage and consent), we compute mothers' empirical transition probabilities. In discussion with ARMMAN, we define 4 groups based on mothers' education, income, and phone ownership status (see Table 3 in Appendix B.3.3). These groups are in proportions [26%, 38%, 29%, 7%], and Group C and D need more resources. Instead of giving service calls to mothers who respond better to service calls, ARMMAN aims to not leave out groups who do not respond as well. Finally, using mothers' empirical transition probabilities and group mapping, we run a simulated RMAB experiment for a subpopulation with $N = 10000$ and $B = 300$.

## 5.1 EXPERIMENTAL RESULTS

We consider two Fairness objectives: *Max Nash-Welfare (MNW)* and *Maximin Reward (MMR)*. We report the respective Fairness Objective and the *Utility*, which is the total sum of rewards of all arms. We report the fairness metrics across 20 different seeds. For each seed, we generate 10 instances of RMAB problems for each setting and split the instances into 70% training, 10% validation, and 20% testing sets. Since every seed value results in different best and worst results, we use a popular min-max normalization Henderi et al. [2021], Gajera et al. [2016]

In Figure 1, we show results on synthetic data with different numbers of groups and fairness objectives while in Figure 2, we show results from real-world health information data experiments. Across all problem scenarios, we observe that DF-LearnB outperforms baselines in terms of fairness metric, achieving 20-40% gains (see Tables 1 and 2 in Appendix for more detailed comparison). This demonstrates the advantage of differentiating through the budget allocation in network updates, and the need for our fully differentiable pipeline. When optimizing for MNW, DF-LearnB achieves utility comparable to DF baselines, including a state-of-the-art DF algorithm [Wang et al., 2023] that purely optimizes utility. When optimizing for MMR, we see a much higher gain in fairness but at the expense of a noticeable drop in Utility (see Figure 1(c-d,g-h)). This suggests that different fairness metrics have varying levels of fairness-utility trade-offs. While DF-LearnB can effectively optimize for the desired objective, the utility cost depends on the objective and so the choice of the objective can be application-dependent. In Appendix B, we present additional results showing similar trends across different values of $N \in \{100, 200, 500\}$ and $B \in \{0.2N, 0.4N, 0.6N\}$.

**Learning Budget Allocation** To further analyze how DF-LearnB produces high fairness objective values, we showcase the allocation of budget across training epochs. Figure 3 illustrates results on **Synthetic** with MNW objective

(see Appendix B.2 Figure 8 for results on MMR). While DF-LearnB slowly adds up the budget to the disadvantaged group alongside simultaneous learning of transition probabilities, DF-GreedyB has unstable updates and fails to allocate sufficient resources to the disadvantaged group. This demonstrates the importance of a fully differentiable pipeline. The budget allocation of Killian et al. [2023] is completely separated from the learning of transition probabilities, which thus leads to suboptimal budget allocation. Finally, the proportional group allocation is a fixed strategy allocating 50% of the budget to disadvantaged group while the DF-NoFair strategy completely starves the disadvantaged group of all resources. We make similar observations on **real-world ARMMAN data**. In Figure 4, we show the budget distribution when the MMR objective is used (see Appendix B.3.4 Figure 8 for corresponding results on MNW). Notice group sizes are unequal in this real-world dataset, as reflected in DF-PropB. Due to the nature of MMR objective, ideal budget allocations should be fully oriented towards groups that are worst off and improve their situation as much as possible. Here Group C and D needs more resources. Notice our DF-LearnB method prioritizes these groups more in need, while other DF baselines fail to provide sufficient resources to these groups.

**Decision Focused Evaluation** To evaluate how close a method is to an optimal top-k selection, we analyze the rank correlation between Whittle indices computed from ground truth transition probabilities and these computed from learned probabilities. Since the top-k selection is performed for every group, we compute rank correlations at a group level and report the Average Group Rank Correlation (AGRC) over all groups and states.

$$\text{AGRC} = \frac{1}{|G||S|} \sum_{s \in S, g \in G} Spearmanr(W_s^g, \hat{W}_s^g) \quad (8)$$

where $W_s^g$ and $\hat{W}_s^g$ are respectively the true and predicted Whittle index lists for group $g$ in state $s$. *Spearmanr* is the Spearman's rank coefficient.

We find that for synthetic data experiment optimizing MNW metric with $|G| = 2$, AGRC is 0.114 for Killian et al. [2023] and is 0.307 for our DF-LearnB. The low correlation values indicate that it is a hard problem to retrieve true transition probabilities and thus true Whittle index ordering for all arms within a group. However, a larger AGRC value for DF-LearnB clearly shows that DF-LearnB learns a better ordering of arms as compared to Killian et al. [2023], resulting in decisions that yield higher fairness objective values.

**Runtime** In Figure 5a we demonstrate that our DF-LearnB has runtime substantially smaller than Killian et al. [2023] and comparable to baselines that ignore fairness. Specifically, DF-LearnB runtime is linear in the number of arms. In Figure 5b, we show DFL approaches that incorporate fairness have runtime linear in the number of groups, which aligns with our theoretical analysis (see Section 4.2).

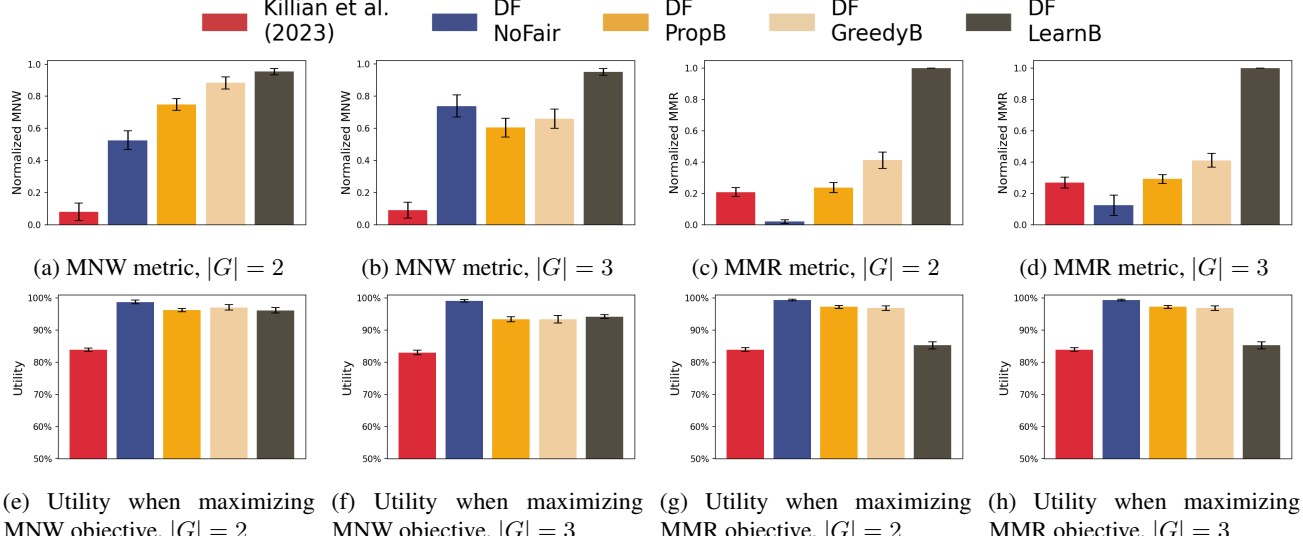

(a) MNW metric, $|G| = 2$    (b) MNW metric, $|G| = 3$    (c) MMR metric, $|G| = 2$    (d) MMR metric, $|G| = 3$

(e) Utility when maximizing MNW objective, $|G| = 2$    (f) Utility when maximizing MNW objective, $|G| = 3$    (g) Utility when maximizing MMR objective, $|G| = 2$    (h) Utility when maximizing MMR objective, $|G| = 3$

Figure 1: We report the min-max normalized Henderi et al. [2021] fair objective metric (a-d) and Utility (e-h) across 20 different seeds in synthetic data experiments. Here $N = 200$, $B = 80$ (see Appendix B.2 for more comprehensive results).

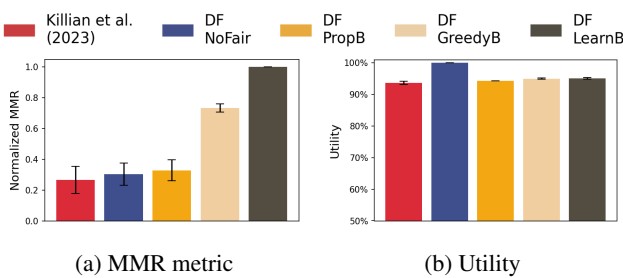

(a) MMR metric      (b) Utility

Figure 2: Fair objective metric and Utility metric when optimizing for MMR objective in real-world data experiments. For all experiments, we have N=10000, B=300 (see Appendix B.3.4 Figure 9 for corresponding results on MNW)

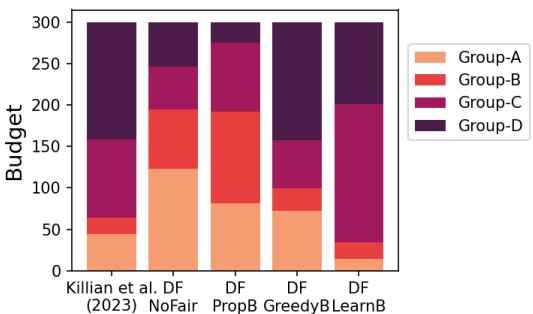

Figure 4: Budget allocation to different risk groups (see Appendix Table 3) in real-world health information data experiment.

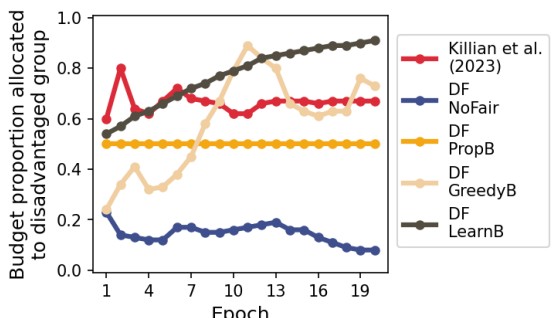

Figure 3: Budget allocation to disadvantaged group, across training epochs in synthetic data experiment with $|G| = 2$.

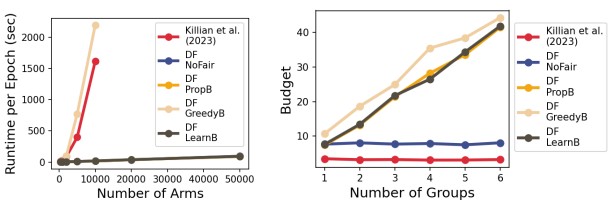

(a) Runtime per epoch with changing number of arms    (b) Runtime per epoch with changing number of groups

Figure 5: Runtime comparison of different algorithms

## 6 CONCLUSION

We provide a novel decision-focused-learning pipeline for equitable RMABs, to prevent disparity between groups. Our

algorithm simultaneously learns transition probabilities and per-group budget allocation. We propose a novel differentiable budget allocation method and provide theoretical results that shed light on the feasibility and effectiveness of this method. Notably, our techniques including the budget allocation method and the differentiable pipeline can be used

to incorporate various fairness notions such as Max Nash Welfare, Maximin, and Gini Index. Our empirical results on both synthetic and real-world large-scale RMAB problems demonstrate that our method significantly improves performance as measured in an equitable objective, with little sacrifice in utility.

# 7 ACKNOWLEDGEMENTS

This work was partially supported by the Harvard Data Science Initiative, ARO (W911NF-18-1-0208), and DARPA (HR001122C0182). The views and conclusions contained in this document are those of the authors and should not be interpreted as representing the official policies, either expressed or implied, of ARO, DARPA, or the U.S. Government. The U.S. Government is authorized to reproduce and distribute reprints for Government purposes notwithstanding any copyright notation herein.

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

# Group Fairness in Predict-Then-Optimize Settings for Restless Bandits (Supplementary Material)

**Shresth Verma**[*1]  **Yunfan Zhao**[*1]  **Sanket Shah**[1]  **Niclas Boehmer**[1]  **Aparna Taneja**[2]  **Milind Tambe**[1,2]

[1]Harvard University
[2]Google Research India

## A    ADDITIONAL ALGORITHM DETAILS

Figure 6 provides an overview of the pipeline, illustrating the difference between two-stage training and decision-focused-learning.

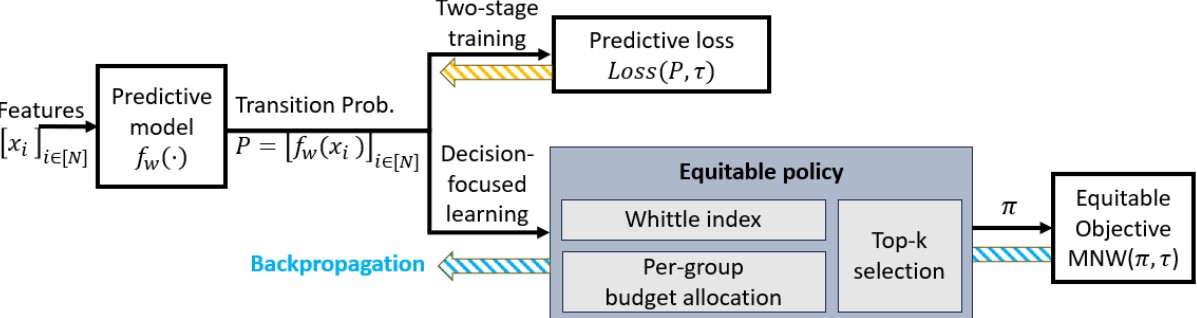

Figure 6: This figure illustrates different methods to train the predictive model $f_w(\cdot)$. Two-stage training uses the predictive loss to perform gradient updates. In contrast, equitable decision-focused-learning backpropagates through the entire pipeline of computing equitable budget allocation and formulating an equitable policy.

### A.1    GRADIENTS WITH RESPECT TO WHITTLE INDICES

We define the value function for an arm $i \in [N]$ as

$$V_i^\lambda(s) = \max_{a \in \{0,1\}} Q_i(s, a_i, \lambda). \tag{9}$$

The value function captures the discounted cumulative reward an arm could receive, assuming all future actions are taken optimally. Recall the Whittle index associated to the state $s_i$ is:

$$W_i(s_i) := \inf_m \left\{ Q_i(s_i, a_i = 0, m) = Q_i(s_i, a_i = 1, m) \right\}.$$

. We obtain for each arm $i \in [N]$ that

$$V_i^\lambda(s) \geq \left\{ \begin{array}{l} \lambda + R(s) + \beta \sum_{s' \in \mathcal{S}} P_i(s, a = 0, s') V_i^\lambda(s') \\ R(s) + \beta \sum_{s' \in \mathcal{S}} P_i(s, a = 1, s') V_i^\lambda(s') \end{array} \right. \tag{10}$$

---

[*]Equation Contribution
[*]Equation Contribution

The above system of inequalities can be rewritten in matrix form [Wang et al., 2023]. Picking only rows where equality holds, we obtain

$$A \begin{bmatrix} \mathbf{1}_M & \beta P_i(\mathcal{S}, a=0, \mathcal{S}) - I_M \\ \mathbf{0}_M & \beta P_i(\mathcal{S}, a=1, \mathcal{S}) - I_M \end{bmatrix} \begin{bmatrix} \lambda \\ V_i^\lambda \end{bmatrix} = -A \begin{bmatrix} r(S) \\ r(S) \end{bmatrix} \tag{11}$$

where the binary matrix $A \in \{0,1\}^{(M+1) \times 2M}$ has row sums all equal to one. The above system of equations is full-rank and its solution is the Whittle index [Wang et al., 2021]. Thus, equation 11 allows us to compute $\frac{dW}{dP}$, the gradient of Whittle indices with respect to transition probability predictions.

## A.2   IMPORTANCE SAMPLING

In offline learning, importance sampling is commonly used to evaluate a target policy distinct from a behavioral policy (or a mixture of behavioral policies) that generates the offline dataset Horvitz and Thompson [1952], Tokdar and Kass [2010]. Specifically, the inverse propensity scores are defined as

$$\frac{\mathbb{I}\{a_i = a\}}{\mu(a_i|s_i)}, \tag{12}$$

where $\mu(a_i|s_i)$ denotes the probability we choose action $a_i$ on state $s_i$, following the behavioral policy.

More generally, given a target policy $\pi(\cdot, \cdot)$, one may use the following version of inverse propensity score:

$$\frac{\pi(a_i|s_i)}{\mu(a_i|s_i)}, \tag{13}$$

Let $\hat{R}(s_i, a)$ denote the reward estimates obtained using inverse propensity scores. Let $\mathcal{A}$ denote the action space. Notably, $\hat{R}(s_i, a)$ is unbiased:

$$\mathbb{E}\left[\hat{R}(s_i, a) \mid s_i, a\right] = \sum_{a' \in \mathcal{A}} \mu(a' \mid s_i) \mathbb{E}\left[\hat{R}(a_i, a) \mid s_i, a, a_i = a'\right] = \sum_{a' \in \mathcal{A}} \mu(a' \mid s_i) r(s_i, a') \frac{\mathbf{1}(a' = a)}{\mu(a' \mid s_i)} = r(s_i, a).$$

For simplicity, in the above derivations we used the simpler notion of inverse propensity scores (12), and we assume that reward observations $r(s_i, a)$ are without noise. Similar results of unbiasedness can be obtained for the more general notion (13) Tokdar and Kass [2010]. When there is noise in reward observations, one may substitute $r(\cdot, \cdot)$ above by the expected reward and obtain a similar result.

## A.3   THE GINI INDEX OBJECTIVE

One could replace MaxNashWelfare (MNW) objective in Algorithm 2 and 3 with the Gini Index objective

$$\frac{\sum_{i=1}^{N} \sum_{j=1}^{N} |x_i - x_j|}{2N^2 \bar{x}},$$

where $\bar{x} = \frac{1}{N} \sum_i x_i$. In implementation, the absolute value can be replaced by $\max\{x_i - x_j, x_j - x_i\}$. For equity in RMABs, we apply the objective on group values $V(s_g, b_g)$.

# B   ADDITIONAL EXPERIMENTAL DETAILS AND RESULTS

## B.1   IMPLEMENTATION AND HYPERPARAMETERS

When implementing the gradient computations described in Proposition 1 proof, we used the Python optimal transport library [Flamary et al., 2021]. For soft top-k selection, we use $\epsilon = 0.01$. We consider a feature dimension of 16 which are correlated with the transition probabilities. For learning the mapping between features and transition probabilities, we use a 3 layer fully connected neural network and add dropouts to prevent overfitting. We use learning rate lr $= 0.001$ for updating the weights of the neural network. We run all experiments with time horizon $H = 10$ and discount factor $\gamma = 0.99$. We run synthetic experiments for 20 epochs and ARMMAN experiments for 30 epochs.

Table 1: A comparison of MaxNashWelfare objective for different methods, under different number of arms $N$, budget $G$. The results illustrate that our DF-LearnB algorithm consistently outperforms baselines, often achieving 20-40% gains in performance.

| N | \|G\| | B | DF-GreedyB | DF-LearnB | DF-NoFair | DF-PropB | Killian et al. (2023) |
|---|---|---|---|---|---|---|---|
| 100 | 2 | 20 | 1844.4395±26.58 | **2075.7222±14.98** | 1855.7535±19.28 | 1964.265±13.66 | 1509.0015±27.26 |
| | | 40 | 3099.4233±31.0 | **3201.858±32.83** | 2848.6445±35.56 | 2967.8445±24.53 | 2554.0105±22.52 |
| | | 60 | 4331.004±70.77 | **4899.134±35.32** | 4543.9746±79.88 | 4311.2197±41.29 | 3781.4805±30.23 |
| | 3 | 20 | 9772.077±180.51 | **10345.396±113.55** | 9554.029±136.11 | 9294.717±70.64 | 7410.884±262.82 |
| | | 40 | 18594.375±332.59 | 20877.344±350.93 | **20877.727±322.38** | 17690.48±244.01 | 14015.834±291.23 |
| | | 60 | 30537.754±518.93 | 33922.445±639.43 | **34160.22±628.87** | 30166.115±487.45 | 24601.406±494.18 |
| 200 | 2 | 40 | 7598.952±94.18 | **8109.785±56.59** | 7457.035±73.44 | 7907.203±56.18 | 6322.539±94.12 |
| | | 80 | 6121.2±982.21 | **6416.3003±1029.48** | 4779.436±939.67 | 6033.884±967.59 | 5080.54±815.24 |
| | | 120 | 17639.025±133.53 | **19646.81±176.58** | 18649.248±219.88 | 17160.963±119.82 | 15324.928±141.23 |
| | 3 | 40 | 76300.6±1025.35 | **82425.66±818.74** | 77052.31±885.67 | 77905.2±841.42 | 59414.332±675.85 |
| | | 80 | 73718.67±11895.45 | **85024.91±13699.44** | 61591.492±12132.29 | 71157.48±11506.2 | 56691.207±9164.64 |
| | | 120 | 241364.95±4926.81 | **284413.8±4135.75** | 255120.45±4803.43 | 243100.05±2427.86 | 195661.33±1573.02 |
| 500 | 2 | 100 | 47710.535±335.0 | **51242.055±146.56** | 46619.305±271.52 | 49934.93±210.22 | 39057.38±267.54 |
| | | 200 | 79310.58±571.92 | **81587.65±462.29** | 73139.51±731.59 | 76560.4±609.23 | 63439.438±293.03 |
| | | 300 | 111593.96±824.02 | **122485.984±634.14** | 111663.516±1183.26 | 107888.336±606.95 | 94599.58±286.63 |
| | 3 | 100 | 1185198.8±9542.08 | **1283090.5±7886.83** | 1173831.1±7249.73 | 1235366.4±9119.11 | 919897.7±14505.66 |
| | | 200 | 2316630.8±22941.54 | **2569942.0±28927.02** | 2473116.5±17234.65 | 2297424.5±25828.66 | 1742517.0±12990.4 |
| | | 300 | 3892794.8±41722.41 | **4434235.0±40364.05** | 4183696.0±50922.18 | 3855896.0±20056.38 | 3067652.0±17616.07 |

## B.2 SYNTHETIC DATASET

Here we provide a more comprehensive set of results, and we report the mean and the standard deviation of fairness objectives for all methods in Tables 1 and 2. We observe that our proposed method DF-LearnB substantially outperforms baselines, including both two-stage methods and decision-focused-learning methods.

In synthetic experiments, we set up a disadvantaged group such that the benefit from pulling over not pulling an arm is strictly lower for the disadvantaged group as compared to the advantaged group, and if we do not act on any arms, then the disadvantaged group obtains much lower reward than the advantaged group. Hence in this setting, a method purely maximizing utility would favor arms from the advantaged group because they have higher whittle indices. However, this would induce a high fairness-penalty because not giving interventions to disadvantaged group creates high inequality across groups.

Table 2: A comparison of Maximin Reward objective for different methods, under different number of arms $N$, budget $G$. The results illustrate that our DF-LearnB algorithm consistently outperforms baselines, often achieving 20-40% gains in performance.

| N | \|G\| | K | DF-GreedyB | DF-LearnB | DF-NoFair | DF-PropB | Killian et al. (2023) |
|---|---|---|---|---|---|---|---|
| 100 | 2 | 20 | 94.25±2.14 | **102.5±0.76** | 66.92±0.56 | 88.17±0.85 | 74.33±1.45 |
| | | 40 | 103.67±4.17 | **145.17±1.15** | 96.0±2.32 | 108.25±0.55 | 108.17±0.79 |
| | | 60 | 137.75±4.24 | **171.58±1.02** | 164.75±1.72 | 139.25±1.73 | 133.75±1.43 |
| | 3 | 20 | 55.61±1.15 | **78.78±1.05** | 51.22±0.86 | 61.78±0.61 | 61.39±1.26 |
| | | 40 | 84.33±2.82 | **102.72±1.18** | 65.17±1.07 | 78.94±0.95 | 73.89±1.91 |
| | | 60 | 94.72±2.72 | **114.06±1.24** | 107.61±0.94 | 96.5±1.53 | 87.0±2.1 |
| 200 | 2 | 40 | 191.83±1.39 | **205.17±2.36** | 140.17±1.82 | 177.33±1.5 | 156.83±2.52 |
| | | 80 | 237.12±5.51 | **294.42±2.37** | 197.96±5.0 | 221.38±1.8 | 218.19±2.59 |
| | | 120 | 297.67±5.23 | **350.67±3.19** | 336.83±3.07 | 287.33±3.57 | 279.83±2.6 |
| | 3 | 40 | 112.67±1.33 | **156.56±1.68** | 96.44±1.04 | 120.89±1.57 | 120.44±1.32 |
| | | 80 | 162.62±4.27 | **214.69±1.71** | 134.95±6.53 | 153.31±2.01 | 151.49±2.65 |
| | | 120 | 209.56±3.5 | **236.89±3.18** | 228.44±2.5 | 189.44±2.56 | 176.0±2.19 |
| 500 | 2 | 100 | 495.83±2.05 | **527.08±2.04** | 340.83±2.57 | 445.0±2.09 | 395.0±3.5 |
| | | 200 | 647.08±7.77 | **740.0±1.73** | 476.25±13.8 | 550.83±3.27 | 550.83±4.63 |
| | | 300 | 749.58±3.96 | **863.75±3.06** | 830.0±2.89 | 712.92±2.82 | 657.08±3.23 |
| | 3 | 100 | 327.5±4.22 | **406.67±3.18** | 239.44±1.87 | 303.06±3.03 | 307.78±3.14 |
| | | 200 | 418.89±11.0 | **548.33±3.18** | 303.61±2.01 | 381.39±1.71 | 384.72±2.19 |
| | | 300 | 525.0±5.39 | **596.67±2.04** | 579.44±2.31 | 488.61±2.99 | 438.61±3.13 |

### B.3 REAL-WORLD ARMMAN DATASET

#### B.3.1 Secondary Analysis

Our experiment falls into the category of secondary analysis of the data shared by ARMMAN. This paper does not involve the deployment of the proposed algorithm or any other baselines to the service call program. As noted earlier, the experiments are secondary analysis with approval from the ARMMAN ethics board.

#### B.3.2 Consent and Data Usage

Consent is obtained from every beneficiary enrolling in the NGO's mobile health program. The data collected through the program is owned by the NGO and only the NGO is allowed to share data. In our experiments, we use anonymized call listenership logs to calculate empirical transition probabilities. No personally identifiable information (PII) is available to us. The data exchange and usage were regulated by clearly defined exchange protocols including anonymization, read-access only to researchers, restricted use of the data for research purposes only, and approval by ARMMAN's ethics review committee.

#### B.3.3 Risk Attributes

Table 3 shows the risk attributes in the real-world ARMMAN dataset. We also show the percentile distribution of the Whittle Index for every group in Figure 7.

| Risk Attribute | Definition | Population Proportion |
|---|---|---|
| Low Income | Monthly Family Income <INR 15,000 (180 USD approx) | 55.5% |
| Low Education | Highest Education Level matriculation or below | 37.2% |
| Phone Owner | Phone not owned by beneficiary | 23.1% |

Table 3: Risk attributes, their definitions, and their prevalence in the real-world data. Each risk attribute contributes to the risk score by 1 resulting in a risk score value between 0 and 3

Group A has mothers who do not have any of the risk attributes. Group B has mothers with 1 risk attribute, Group C with 2 risk attributes, and Group D with all 3 risk attributes.

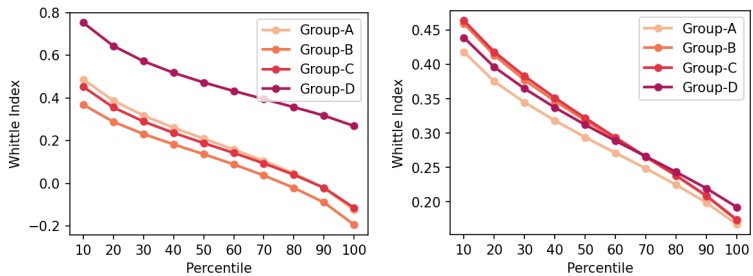

(a) Whittle Index for intervening in Non-Engaging State

(b) Whittle Index for intervening in Engaging State

Figure 7: Whittle Index for different risk groups over top-k percentile

### B.3.4 Group Sizes

In the main paper, we presented results on ARMMAN with unequal group sizes. The group sizes are unequal since the proportion of people in distinct groups in the real-world are different. It is known that Max Nash Welfare favors groups of smaller sizes and may not be blinded used on equal group size settings [Killian et al., 2023]. Thus, for results on MNW objective, we sample a subset of the population such that the group sizes are equal. We present results on equal group sizes in Figure 8 and 9.

When MNW objective is used, we observe that DF-NoFair allocates nearly all budgets to Group-D, whose mothers have much higher Whittle indices (see Figure 7). Killian et al. [2023] and DF-GreedyB allocate budgets close to proportional budgets. While baselines give extreme budget allocations (either all to group-D or simply equal allocations), our DF-LearnB balances fairness and utility and prioritizes groups more in need.

Similar to our observations in the main paper, in fairness objectives, our DF-LearnB outperforms baselines by a wide margin (see Figure 9). Notice our DF-LearnB greatly improves fairness objective values while achieving utility close to that of a State-of-the-Art algorithm purely maximizing utility (DF-NoFair) [Wang et al., 2023].

Note our pipeline can be used with objectives such as Maximin, which naturally accommodates unequal group sizes. Even for MNW, our methods could be extended to accommodate unequal group sizes by adapting existing techniques, such as resampling arms, to balance group sizes [Killian et al., 2023].

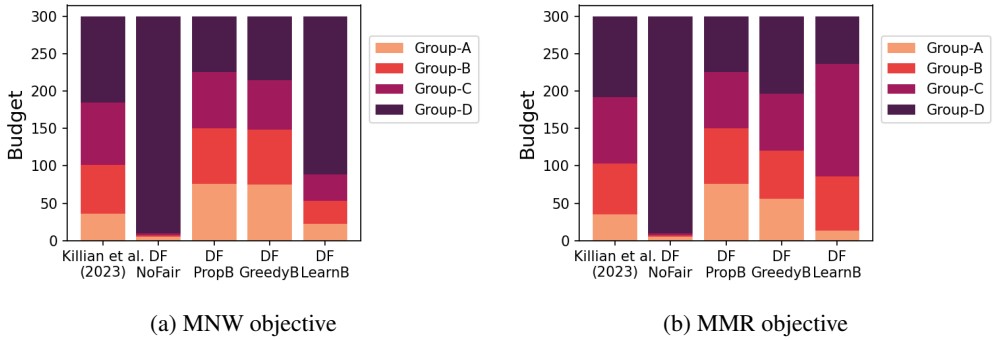

(a) MNW objective          (b) MMR objective

Figure 8: Budget allocation to different risk groups in real-world ARMMAN data experiment with equal group sizes.

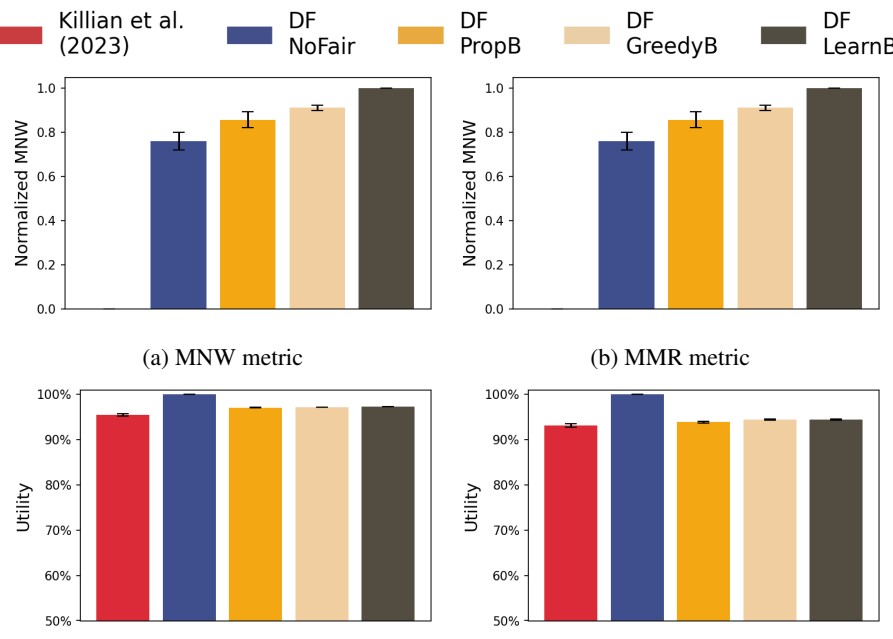

(a) MNW metric

(b) MMR metric

(c) Utility when maximizing MNW objective (d) Utility when maximizing MMR objective

Figure 9: Fair objective metric (a-b) and Utility metric (c-d) for different methods in real-world data experiments. For all experiments, we have N=10000, B=300

## C  PROOF OF THEOREM 1

*Proof.* When there are only two groups $g_1, g_2$, from the budget constraint we have $b_{g_1} = B - b_{g_2}$. If we can show $\hat{h}_{g_1}^{MNW}(\cdot)$ is concave, then by symmetry, the same argument can be applied to $g_2$ to show that $\hat{h}_{g_2}^{MNW}(\cdot)$ is concave.

When there are three or more groups, we can view $\hat{g} = \mathcal{G} \setminus g_1$ as one artificial group. By the assumption that $V_g(b_g)$ has diminishing returns in $b_g$ for any $g \in \mathcal{G}$, we have that $V_{\hat{g}}(b_{\hat{g}})$ has diminishing returns. Consequently, the same argument in the two group case applies. By symmetry, the same argument can be applied to any group other than $g_1$.

Thus, it suffices to show $\hat{h}_{g_1}^{MNW}(\cdot)$ is concave in the two group case.

Since the initial states $s$ are given and fixed, for ease of notation, we drop the dependence of $V_g(s_g, b_g)$ on $s_g$ and write as $V_g(b_g)$.

We will prove by contradiction. Suppose to the contrary that $\hat{h}_{g_1}^{MNW}(\cdot)$ is non-concave. By definition of concave functions, there must exist values $0 \leq x_1 < x_2 \leq B$ and $\alpha \in [0, 1]$ such that

$$\alpha \cdot \hat{h}_{g_1}^{MNW}(x_1) + (1 - \alpha) \cdot \hat{h}_{g_1}^{MNW}(x_2) > \hat{h}_{g_1}^{MNW}(\alpha x_1 + (1 - \alpha)x_2).$$

Since $\hat{h}_{g_1}^{MNW}(\cdot)$ is a piecewise linear function and is a linear extrapolation based on nearest integer points, there must exist an integer $k \in \{1, ..., B - 1\}$ such that

$$\frac{1}{2}h_{g_1}^{MNW}(k - 1) + \frac{1}{2}h_{g_1}^{MNW}(k + 1) > h_{g_1}^{MNW}(k). \tag{14}$$

Figure 10 illustrates the landscape.

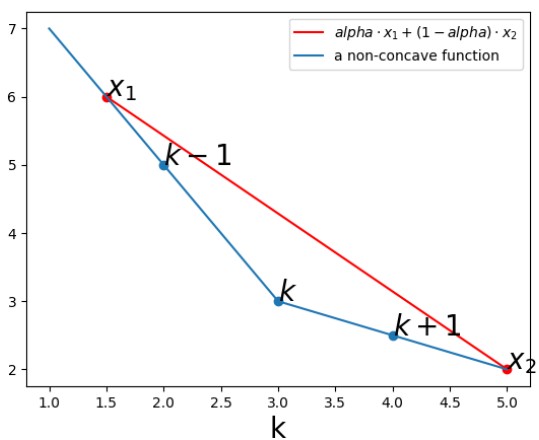

Figure 10: Illustration of a non-concave function in $b_{g_1}$

Multiplying both sides of Equation 14 by 2 and rearranging terms, we obtain:

$$h_{g_1}^{MNW}(k - 1) - h_{g_1}^{MNW}(k) > h_{g_1}^{MNW}(k) - h_{g_1}^{MNW}(k + 1) \tag{15}$$

We consider two cases : (a) $h_{g_1}^{MNW}(k - 1) - h_{g_1}^{MNW}(k) \geq 0$; (b) $h_{g_1}^{MNW}(k - 1) - h_{g_1}^{MNW}(k) < 0$.

**case (a)**

Since $h_{g_1}^{MNW}(k - 1) - h_{g_1}^{MNW}(k) \geq 0$, we have $\epsilon := h_{g_1}^{MNW}(k - 1) - h_{g_1}^{MNW}(k) \geq 0$. By definition of MNW, we can write $h_{g_1}^{MNW}(b_{g_1}) = \log V_{g_1}(b_{g_1}) + \log V_{g_2}(B - b_{g_1})$. Thus,

$$\log V_{g_1}(k - 1) + \log V_{g_2}(B - k + 1) + \epsilon$$
$$= \log V_{g_1}(k) + \log V_{g_2}(B - K)$$

Rearranging terms, we have

$$\log V_{g_1}(k) - \log V_{g_1}(k-1) + \epsilon$$
$$= \log V_{g_2}(B-k+1) - \log V_{g_2}(B-k),$$

Together with the assumption that each group's value function $V_g(b_g)$ has diminishing returns in $b_g$, we have

$$\log V_{g_1}(k+1) - \log V_{g_1}(k) + \epsilon$$
$$\leq \log V_{g_1}(k) - \log V_{g_1}(k-1) + \epsilon$$
$$= \log V_{g_2}(B-k+1) - \log V_{g_2}(B-k)$$
$$\leq \log V_{g_2}(B-k) - \log V_{g_2}(B-k-1)$$

Taking the first and the last line above and rearranging terms, we obtain

$$\log V_{g_1}(k+1) + \log V_{g_2}(B-k-1) + \epsilon$$
$$\leq \log V_{g_1}(k) + \log V_{g_2}(B-k).$$

Thus, $h_{g_1}^{MNW}(k+1) + \epsilon \leq h_{g_1}^{MNW}(k)$. Consequently, $h_{g_1}^{MNW}(k) - h_{g_1}^{MNW}(k+1) \geq \epsilon = h_{g_1}^{MNW}(k-1) - h_{g_1}^{MNW}(k)$, contradicting Equation 15.

**case (b)**

By Equation 15 and that $h_{g_1}^{MNW}(k-1) - h_{g_1}^{MNW}(k) < 0$, we have

$$h_{g_1}^{MNW}(k) - h_{g_1}^{MNW}(k+1) < h_{g_1}^{MNW}(k-1) - h_{g_1}^{MNW}(k) < 0.$$

Let $\epsilon := h_{g_1}^{MNW}(k+1) - h_{g_1}^{MNW}(k) > 0$. By definition of MNW, we can write $h_{g_1}^{MNW}(b_{g_1}) = \log V_{g_1}(b_{g_1}) + \log V_{g_2}(B - b_{g_1})$. Thus,

$$\log V_{g_1}(k+1) + \log V_{g_2}(B-k-1)$$
$$= \log V_{g_1}(k) + \log V_{g_2}(B-k) + \epsilon$$

Rearranging terms, we have

$$\log V_{g_1}(k+1) - \log V_{g_1}(k)$$
$$= \log V_{g_2}(B-k) - \log V_{g_2}(B-k-1) + \epsilon,$$

Together with the assumption that each group's value function $V_g(b_g)$ has diminishing returns in $b_g$, we have

$$\log V_{g_1}(k) - \log V_{g_1}(k-1)$$
$$\geq \log V_{g_1}(k+1) - \log V_{g_1}(k)$$
$$= \log V_{g_2}(B-k) - \log V_{g_2}(B-k-1) + \epsilon$$
$$\geq \log V_{g_2}(B-k+1) - \log V_{g_2}(B-k) + \epsilon$$

Taking the first and the last line above and rearranging terms, we obtain

$$\log V_{g_1}(k) + \log V_{g_2}(B-k)$$
$$\geq \log V_{g_1}(k-1) + \log V_{g_2}(B-k+1) + \epsilon.$$

Thus, $h_{g_1}^{MNW}(k) \geq h_{g_1}^{MNW}(k-1) + \epsilon$. Consequently, $h_{g_1}^{MNW}(k) - h_{g_1}^{MNW}(k-1) \geq \epsilon = h_{g_1}^{MNW}(k+1) - h_{g_1}^{MNW}(k)$, contradicting Equation 15.

Thus, in either case (a) and case (b), we have shown there is a contradiction. We conclude that $\hat{h}_g^{MNW}(\cdot)$ is concave for any group $g \in \mathcal{G}$.

$\square$

# D PROOF OF THEOREM 2

*Proof.* We start with defining linear extrapolations of group value functions:

$$\hat{V}_g(b_g) := V_g(\lfloor b_g \rfloor) + (b_g - \lfloor b_g \rfloor) \cdot (V_g(\lceil b_g \rceil) - V_g(\lfloor b_g \rfloor)), \quad \forall g$$

Observe that $\hat{V}_g(b_g) = V_g(b_g)$ for $b_g \in \mathbb{Z}$ and on non-integer valued $b_g$ the function $\hat{V}_g(b_g)$ is a linear extrapolation based on nearest integer points. By the assumption that $V_g(b_g)$ has diminishing returns in $b_g$ for any $g \in \mathcal{G}$, we have that $\hat{V}_g(b_g)$ is concave in $b_g$.

When there are only two groups, $b_{g_1} = B - b_{g_2}$. Since $\hat{V}_{g_2}(\cdot)$ is concave, we have that $\hat{V}_{g_2}(B - b_{g_1})$ is a concave function of $b_{g_1}$. Since the minimum of concave functions is concave, we have that $\min(\hat{V}_{g_1}(b_{g_1}), \hat{V}_{g_2}(B - b_{g_1}))$ is a concave function of $b_{g_1}$. Thus, $\hat{h}_{g_1}^{MMR}(\cdot)$ is concave. By symmetry, the same argument can be applied to $b_{g_2}$ to show that $\hat{h}_{g_2}^{MMR}(\cdot)$ is concave. Thus, we proved the statement in the two groups case.

When there are three or more groups, we can view $\hat{g} = \mathcal{G} \setminus g_1$ as one artificial group. By the assumption that $V_g(b_g)$ has diminishing returns in $b_g$ for any $g \in \mathcal{G}$, we have that $V_{\hat{g}}(b_{\hat{g}})$ has diminishing returns. Consequently, the same argument in the two group case applies. $\qquad\square$