# OpenReview forum: "Group Fairness in Predict-Then-Optimize Settings for Restless Bandits"
_auai.org/UAI/2024/Conference — UAI 2024 oral_

### Official Review · Reviewer_a3Ma · 2024-02-28

**Q2-1 Originality-Novelty:** 2
**Q2-2 Correctness-Technical Quality:** 3
**Q2-5 Clarity Of Writing:** 3

**Q1 Summary And Contributions:**

This draft tries to study Group Fairness in Predict-Then-Optimize Settings in Restless Multi-armed Bandits where groups of agents are similar to clusters, that needs to simultaneously learn transition probabilities relates to Collaborative Filtering, both artificial and some real-world data are tested

**Q2-3 Extent To Which Claims Are Supported By Evidence:**

2: Fair: the main claims are somewhat supported by evidence (but the experimental evaluation may be weak, or does not match entirely with the claims, important baselines may be missing, proofs contain important ideas but lack rigor, algorithmic details are only discussed superficially, references are imprecise, assumptions are not sufficiently motivated or explicated, etc.).

**Q2-4 Reproducibility:**

3: Good: key resources (e.g. proofs, code, data) are available and key details (e.g. proofs, experimental setup) are sufficiently well-described for competent researchers to confidently reproduce the main results.

**Q3 Main Strengths:**

1. this draft is easy to understand and follow
2. your core idea of budget allocation algorithm to prevent disparity between groups is resonable

**Q4 Main Weakness:**

1. your emprical performance is not impressive, and how this proposal is useful in practice or industry is unclear, which should be clearly articulated
2. in your experiments, only 10k of N is a bit small where you don't have large scaled and real-world or production data based experimental results to support your claims which is one of the main drawbacks of this work

**Q5 Detailed Comments To The Authors:**

This manuscript studies decision-focused-learning pipeline for equitable RMabs among groups, with the potential to combine several fairness notions, there are related state-of-the-art which are sharing the similar spirit that you may want to compare: Improved Algorithm on Online Clustering of Bandits, Collaborative Filtering Bandits, Fast Distributed Bandits for Online Recommendation Systems

**Q9 Complying With Reviewing Instructions:**

Yes

---

> ### Author Rebuttal · Authors · 2024-04-04
>
> We sincerely thank the reviewer for the valuable comments.
>
> “how this proposal is useful in practice or industry is unclear, which should be clearly articulated”
>
> A: Our empirical results on a dataset collected in a real-world public health setting demonstrate that our method is scalable and provides better fairness results than currently used algorithms, with little sacrifice in total utility (summed social welfare);  see Figures 1 & 2, thus showcasing the practicality of our approach.
>
> “in your experiments, only 10k of N is a bit small where you don't have large scaled and real-world or production data based experimental results to support your claims which is one of the main drawbacks of this work”
>
> A: Existing works on restless bandits mostly focus on modeling limited resource allocation problems with 50 to 10k arms [Wang et al. 2023; Akbarzadeh and Mahajan 2019; Li and Varakantham 2022]. Our complexity analysis (end of Sec 4.3) and runtime comparison (Figure 5a) show that our main algorithm (DF-LearnB) has a runtime approximately linear in the number of arms and therefore is in fact more scalable than existing approaches for group fairness in RMABs, whose runtime has a O(N log(N)) dependence on the number N of arms (Killian et al. [2023]). We will further elaborate on the scale of our experiments in the camera ready version.
>
> “there are related state-of-the-art which are sharing the similar spirit that you may want to compare: Improved Algorithm on Online Clustering of Bandits, Collaborative Filtering Bandits, Fast Distributed Bandits for Online Recommendation Systems”
>
> A: We thank the reviewer for pointing us to the relevant works. These works are in online learning, whereas we consider an offline learning problem. In high-stake resource allocation problems where ensuring fairness is critical, e.g., the mobile health programs, it is common that the time horizon is very small (for the considered ARMMAN dataset only 12 timesteps / weeks of data are used), and thus online learning would be impractical. We will include a discussion on these relevant works in the camera ready version.

---

### Official Review · Reviewer_VhC7 · 2024-03-13

**Q2-1 Originality-Novelty:** 2
**Q2-2 Correctness-Technical Quality:** 3
**Q2-5 Clarity Of Writing:** 3

**Q1 Summary And Contributions:**

This paper presents a method to solve the resource allocation problem with group fairness budget constraints. The authors use a differentiable budget allocation procedure to gradually update the budget allocations. They evaluate the fairness objectives such as MNW, MMR in the proposed pipeline. Experiments showcase good results on both synthetic and real-world datasets.

**Q2-3 Extent To Which Claims Are Supported By Evidence:**

3: Good: the main claims are supported by convincing evidence (in the form of adequate experimental evaluation, proofs, (pseudo-)code, references, assumptions).

**Q2-4 Reproducibility:**

3: Good: key resources (e.g. proofs, code, data) are available and key details (e.g. proofs, experimental setup) are sufficiently well-described for competent researchers to confidently reproduce the main results.

**Q3 Main Strengths:**

The paper is well-motivated and clearly highlights the limitations of current works.

Sound experimental results and evaluations clearly show the proposed framework's improvements over existing baselines.

**Q4 Main Weakness:**

While the structure of the paper is largely lucid, some parts lack clear explanation. Section 4.x requires further refinement, for example, adding arguments for the design choices about why using a neural network to learn the transition matrix. It would be beneficial to elaborate on these aspects rather than merely mentioning them. For more detailed questions, see Q5.

**Q5 Detailed Comments To The Authors:**

* Algorithm 1 is essentially equal to the greedy algorithm using Whittle index-based approach: firstly rank all arms according to the whittle index value, then select the arms to satisfy the group budget constraint, followed by adding the extra top-ranked arms to meet the overall budget. The concept of group or community fairness constraints has been investigated by other works. To my understanding, the above approach is deemed efficient without sacrificing a lot performances for resource allocation with budget constraints if the RMAB is indexable.

* Regarding the requirement for the differentiability in budget allocation, the advantages of necessitating differentiability aren’t apparent to me. Besides, computing $J_{\Delta}(s_g,b_g)$ is easier comparing to compute the gradient (as indicated in the limitations).

* The authors mentioned using the neural network to learn the transition matrix when there is uncertainty in the transition matrix, which would influence the whittle index value. However, there is a lack of explanation regarding why using neural networks, what is the benefits of using neural network compared to the Thompson sampling based learning approach.

* Algorithm 3 involves initializing the budget $b_g$ and gradually updating the budget allocation. However, it's unclear how the authors ensure that the final budget allocation $b_g$ satisfies the final group constraints.

**Q9 Complying With Reviewing Instructions:**

Yes

---

> ### Author Rebuttal · Authors · 2024-04-04
>
> We sincerely thank the reviewer for the valuable comments.
>
> “Algorithm 1 is essentially equal to the greedy algorithm using Whittle index-based approach: firstly rank all arms according to the Whittle index value, then select the arms to satisfy the group budget constraint, followed by adding the extra top-ranked arms to meet the overall budget. The concept of group or community fairness constraints has been investigated by other works. To my understanding, the above approach is deemed efficient without sacrificing a lot performances for resource allocation with budget constraints if the RMAB is indexable.”
>
> A: Indeed, Algorithm 1 is a greedy approach. However, no per-group budgets are given and thus before we pick high Whittle index arms within each group to fill the group budget, a key challenge is to decide on per-group budget allocations.  Although previous works considered fairness and among them Killian et al. [2023] considered group budget allocations, they ignored a main challenge in using RMABs in the real world: agents’ transition probabilities are often unknown and need to be learned.
>
> Thus, the main challenge we address in our work is to simultaneously learn transition probabilities and per-group budget allocations, to optimize pre-defined fairness metrics. Our fully differentiable pipeline incorporates how both transition probabilities and budget allocations affect the fairness objective, and it leads to substantially stronger performance than non-differentiable approaches (see Figure 1 & 2). We will highlight the above points in the camera-ready version.
>
> “Regarding the requirement for the differentiability in budget allocation, the advantages of necessitating differentiability aren’t apparent to me. Besides, computing J_delta(s_g,b_g)  is easier compared to compute the gradient (as indicated in the limitations).”
>
> A: If we understand the reviewer’s question correctly, they are asking about the necessity of a fully differentiable pipeline (Alg 3) when we already have the non-differentiable Alg 1. Figures 1 & 2 show the substantial performance gains of a differentiable pipeline. Alg 1 (DF+GreedyB) takes a greedy, non-differentiable approach, with no gradients with respect to the budget calculated. Alg 1 fails to consider db/dW, i.e., the effect of the changes in budget allocations w.r.t changes in Whittle, and consequently suffers from issues such as: (a) when transition probability predictions are poor, J_\delta(s_g, b_g) estimates are inaccurate, and the resulting policy fails to allocate sufficient resources to disadvantaged groups (Fig 4); (b) because the non-differentiable approach does not take into account db/dW, DF+GreedyB has unstable updates (Fig 3). We will further emphasize these throughout Sec 4.
>
> “The authors mentioned using the neural network to learn the transition matrix when there is uncertainty in the transition matrix, which would influence the whittle index value. However, there is a lack of explanation regarding why using neural networks, what is the benefits of using neural network compared to the Thompson sampling based learning approach.”
>
> A: We consider an offline learning setting where we have access to a “historic” dataset (see first line of Algo 2 and Sec 5 dataset descriptions) containing a set of arms (different from the ones our policy runs on) together with their features and some state-action trajectory. We will further clarify the setting in the camera ready version. We use a neural network to utilize feature information and estimate the transition probabilities of new arms from the offline dataset efficiently. A Thompson sampling based approach is more suitable for an online learning problem, where we do not have access to a historic dataset and need to learn the arms’ transition probabilities in real-time. We work in this offline setting because, in high-stakes healthcare settings, it is common that only a few weeks of data are available (for the ARMMAN dataset studied only 12 steps/weeks of data are available), and thus online learning would be impractical.
>
> “Algorithm 3 involves initializing the budget and gradually updating the budget allocation. However, it's unclear how the authors ensure that the final budget allocation satisfies the final group constraints.”
>
> A: The total budget is given and fixed, but a key challenge is to learn the proportions of the total budget to allocate to each group. When performing gradient updates, we project the learned budget onto the feasible region (we apply softmax normalization in the last layer of the neural network learning per group budget allocations). Thus, the budget constraint is satisfied throughout. We will clarify this when introducing Algorithm 3.

---

### Official Review · Reviewer_TXNZ · 2024-03-18

**Q2-1 Originality-Novelty:** 3
**Q2-2 Correctness-Technical Quality:** 3
**Q2-5 Clarity Of Writing:** 4

**Q1 Summary And Contributions:**

The submission provides an algorithm for a restless multi-armed bandit problem that is both (i) decision-focused and (ii) fairness-constrained. If I understand correctly, previous algorithms have been either (i) or (ii) but not both. It develops a “differentiable pipeline” that allows transition probability estimates to be updated in a way that is sensitive to the fairness objective, rather than just predictive accuracy. This has a nice theoretical justification and results in improvements in empirical performance.

**Q2-3 Extent To Which Claims Are Supported By Evidence:**

4: Excellent: all claims are supported by very convincing evidence (in the form of comprehensive experimental evaluation, rigorous mathematical proofs, detailed (pseudo-)code, precise references, well-motivated and realistic assumptions) and the authors deliver what they promise.

**Q2-4 Reproducibility:**

4: Excellent: key resources (e.g. proofs, code, data) are available and key details (e.g. proof sketches, experimental setup) are comprehensively described for competent researchers to confidently and easily reproduce the main results.

**Q3 Main Strengths:**

The submission is addressing a very interesting and important problem. It is technically sophisticated, well-motivated and well-written. As far as I can tell, its results are state-of-the-art.

**Q4 Main Weakness:**

The novel contribution of the paper is not so clearly articulated. If I have not made a mistake, I think I have more clearly stated the novelty in my answer to Q1: Wang et al (2023) and Verma (2023a) are decision-focused, but do not incorporate fairness, whereas Killian et al. (2023) incorporates fairness, but is not decision-focused. The proposed algorithm is the only one which is both.

The empirical results compare only to Killian et al. (2023) and other methods that the authors propose. Why not include methods from Verma et al (2023a) and Wang et al. (2023)? Is it because these don’t incorporate fairness? Still, this would be a nice sanity check on your DFNoFair algorithm.  I think you could add a sentence about why you compare only to Killian  et al. (2023).


The description of the ARMMAN program is a little unclear. You are deciding whether to pull arm i (a *human* should call participant i) and the reward is whether or not participant i listens to an *automated* message for more than 30 seconds? Have I got that right? If so the difference between human/automated calls should be made clearer in the text.

**Q5 Detailed Comments To The Authors:**

I think the paper would be improved with minor adjustments addressing the weaknesses I point out in the previous question.

**Q9 Complying With Reviewing Instructions:**

Yes

---

> ### Author Rebuttal · Authors · 2024-04-04
>
> We sincerely thank the reviewer for the insights and comments.
>
> “The novel contribution of the paper is not so clearly articulated. If I have not made a mistake, I think I have more clearly stated the novelty in my answer to Q1: Wang et al (2023) and Verma (2023a) are decision-focused, but do not incorporate fairness, whereas Killian et al. (2023) incorporates fairness, but is not decision-focused. The proposed algorithm is the only one which is both.”
>
> A: Indeed, as the reviewer points out, to the best of our knowledge, we are the first to incorporate fairness in decision-focused learning for restless bandits. We will further clarify this in the camera-ready version.
>
> “The empirical results compare only to Killian et al. (2023) and other methods that the authors propose. Why not include methods from Verma et al (2023a) and Wang et al. (2023)? Is it because these don’t incorporate fairness? Still, this would be a nice sanity check on your DFNoFair algorithm. I think you could add a sentence about why you compare only to Killian et al. (2023).”
>
> A: The approaches described by Verma et al (2023a) and Wang et al. (2023) are in fact equivalent to the DFNoFair algorithm, which is included as a baseline in our empirical results (Figure 1 & 2): They are decision-focused-learning methods that do not incorporate fairness. We will emphasize this connection/equivalence in the experimental section (Sec. 5).
>
> “The description of the ARMMAN program is a little unclear. You are deciding whether to pull arm i (a human should call participant i) and the reward is whether or not participant i listens to an automated message for more than 30 seconds? Have I got that right? If so the difference between human/automated calls should be made clearer in the text.”
>
> A: Yes! As the reviewer points out, we are deciding whether a human calls a participant, and we get a reward each time a participant listens to an automated message for more than 30 seconds. We will further clarify the difference between human and automated calls.

---

### Official Review · Reviewer_9Dsq · 2024-03-26

**Q2-1 Originality-Novelty:** 3
**Q2-2 Correctness-Technical Quality:** 3
**Q2-5 Clarity Of Writing:** 4

**Q1 Summary And Contributions:**

The paper tackles a relevant problem in the responsible computing literature that of ensuring group fairness while considering the restless MAB setting. The decision focused learning framework (DFL) is enhanced by utilizing the fairness constraint and compared with a two stage process of first predicting the transition probabilities and then optimizing. Experiments are shown on both synthetic and real-world datasets to show the benefits of the proposed approach.

Overall: The paper is well-written and easy to follow with good theoretical results as well as strong experiments.

**Q2-3 Extent To Which Claims Are Supported By Evidence:**

4: Excellent: all claims are supported by very convincing evidence (in the form of comprehensive experimental evaluation, rigorous mathematical proofs, detailed (pseudo-)code, precise references, well-motivated and realistic assumptions) and the authors deliver what they promise.

**Q2-4 Reproducibility:**

4: Excellent: key resources (e.g. proofs, code, data) are available and key details (e.g. proof sketches, experimental setup) are comprehensively described for competent researchers to confidently and easily reproduce the main results.

**Q3 Main Strengths:**

(i) They combine the fairness constraints in their differentiable pipeline enabling them to perform end-to-end optimization utilizing the recently proposed top-k operators from the optimal transport literature which have been widely used.
(ii) Properties that fairness measures need to satisfy are also stated and two in particular MNW and MMR are analyzed as well utilized in the experiments.
(iii) Experiments are provided using a variety of settings -- two stage, no fairness, DFL + group size based budgets, DFL + greedy, proposed approach. Various synthetic datasets are explored as well as varying sizes and number of groups.

**Q4 Main Weakness:**

(a) No regret bounds are provided for the approach. It is unclear how optimal the proposed algorithm is compared to the RMABs without fairness settings. Experimentally, this can be done by varying the budget and showing them to be equivalent when budget is high enough.

**Q5 Detailed Comments To The Authors:**

(1) What is the formal notion of regret in this setting?
(2) Also, the notion of fairness here is in terms of allocation of resources and not necessarily that of rewards for the subgroups (outcomes)?

**Q9 Complying With Reviewing Instructions:**

Yes

---

> ### Author Rebuttal · Authors · 2024-04-04
>
> We sincerely thank the reviewer for the valuable comments.
>
> “No regret bounds are provided for the approach. It is unclear how optimal the proposed algorithm is compared to the RMABs without fairness settings. Experimentally, this can be done by varying the budget and showing them to be equivalent when the budget is high enough.” & “What is the formal notion of regret in this setting?”
>
> A: We consider an offline learning setting where we have access to a “historic” dataset  (see first line of Algo 2 and Sec 5 dataset descriptions) containing a set of arms (different from the ones our policy runs on) together with their features and some state-action trajectory. A regret bound analysis is more suitable for an online learning problem, where we do not have access to a historic dataset and need to learn the arms’ transition probabilities and collect data in real time. For a comparison of our method to RMAB algorithms without fairness, we refer to DF-NoFair in Figures 1 & 2 (see also Tables 1 & 2 in Appendix B). We will further clarify the setting in the camera ready version.
>
> “The notion of fairness here is in terms of allocation of resources and not necessarily that of rewards for the subgroups (outcomes)?”
>
> A: We want to clarify that our fairness notions are ultimately concerned with the rewards of the subgroups (see Eq 4 and Eq 5). Although rewards of the subgroups are greatly affected by the per-group allocation of resources, approaches that only focus on the equal allocation of resources are often not sufficient, as equity or balanced *outcomes* are preferred [1]
>
> [1] Hanan Luss. Equitable resource allocation: models, algorithms and applications. John Wiley & Sons, 2012.

---

### Meta-Review · Area_Chair_TPYt · 2024-04-21

The paper presents a novel approach addressing group fairness in restless multi-armed bandit (MAB) settings, introducing a decision-focused learning framework (DFL). It surpasses previous algorithms by integrating fairness constraints into transition probability estimation, offering theoretical robustness and empirical superiority. Experiments on synthetic and real-world datasets validate its effectiveness in achieving fairness objectives such as MNW and MMR, making it a significant contribution to responsible computing literature.

We accept the paper based on the interesting problem formulation and technical novelties, however please incorporate the final suggestions of the reviewers in the camera-ready version as promised: Precisely comparison of the methods to RMAB algorithms, discussion on the scale of our experiments, clarify the offline learning setting with access to a “historic” dataset, and budget setting.